# An unconventional gatekeeper mutation sensitizes inositol hexakisphosphate kinases to an allosteric inhibitor

Tim Aguirre[1,2], Gillian L Dornan[1], Sarah Hostachy[1], Martin Neuenschwander[1], Carola Seyffarth[1], Volker Haucke[1], Anja Schütz[3], Jens Peter von Kries[1], Dorothea Fiedler[1,2]*

[1]Leibniz-Forschungsinstitut für Molekulare Pharmakologie (FMP), Berlin, Germany; [2]Institut für Chemie, Humboldt-Universität zu Berlin, Berlin, Germany; [3]Max-Delbrück-Center for Molecular Medicine in the Helmholtz Association (MDC), Berlin, Germany

**Abstract** Inositol hexakisphosphate kinases (IP6Ks) are emerging as relevant pharmacological targets because a multitude of disease-related phenotypes has been associated with their function. While the development of potent IP6K inhibitors is gaining momentum, a pharmacological tool to distinguish the mammalian isozymes is still lacking. Here, we implemented an analog-sensitive approach for IP6Ks and performed a high-throughput screen to identify suitable lead compounds. The most promising hit, FMP-201300, exhibited high potency and selectivity toward the unique valine gatekeeper mutants of IP6K1 and IP6K2, compared to the respective wild-type (WT) kinases. Biochemical validation experiments revealed an allosteric mechanism of action that was corroborated by hydrogen deuterium exchange mass spectrometry measurements. The latter analysis suggested that displacement of the αC helix, caused by the gatekeeper mutation, facilitates the binding of FMP-201300 to an allosteric pocket adjacent to the ATP-binding site. FMP-201300 therefore serves as a valuable springboard for the further development of compounds that can selectively target the three mammalian IP6Ks; either as analog-sensitive kinase inhibitors or as an allosteric lead compound for the WT kinases.

## eLife assessment

This manuscript describes a **fundamental** strategy for developing isozyme-selective inhibitors of inositol hexakisphosphate kinases. The **compelling** evidence that subtle changes to the gatekeeper position can sensitize the inositol hexakisphosphate kinase mutant to allosteric inhibitors will undoubtedly inspire other analog-sensitive inhibitor studies. This manuscript will be of interest to researchers focusing on kinase regulation and inhibitor design.

## Introduction

Over the last decades, inositol pyrophosphates (PP-InsPs) have emerged as a group of ubiquitous small molecule messengers with pivotal functions in cellular homeostasis, metabolism, and many disease-related signaling events (*Mukherjee et al., 2020*; *Chakraborty, 2018*). The densely charged molecules are interconverted in a fast metabolic cycle regulated by specific kinases and phosphatases (*Figure 1a*). Genetic knockouts of the three mammalian inositol hexakisphosphate kinase (IP6K) isozymes – enzymes involved in PP-InsP biosynthesis – have revealed their involvement in distinct biological processes, including insulin signaling (*Bhandari et al., 2008*; *Manning, 2010*; *Chakraborty*

*For correspondence:
fiedler@fmp-berlin.de

**Figure 1.** The inositol pyrophosphate pathway is a therapeutically relevant target for small molecule inhibitors. (**a**) Simplified inositol pyrophosphate metabolism with a focus on kinases. PPIP5K: inositol hexakisphosphate and diphosphoinositol-pentakisphosphate kinase. *PP-InsP dephosphorylation is catalyzed by nudix hydrolases DIPP1/2α/2β/3 (*Lonetti et al., 2011*; *Hua et al., 2003*; *Saiardi et al., 2000*). (**b**) Chemical structures and properties of selected IP6K inhibitors: widely used TNP, isozyme-selective Barrow-24, and potent inhibitor SC-919.

*et al., 2010*), apoptosis (*Rao et al., 2014*; *Koldobskiy et al., 2010*; *Morrison et al., 2001*), cancer metastasis (*Rao et al., 2015*), and lifespan regulation in mice (*Moritoh et al., 2016*). Despite catalyzing the same biochemical reaction, some IP6K functions were exclusively attributed to one isozyme. For example, knockdown of IP6K1, but not IP6K2, inhibited the exocytosis of insulin-containing granules in pancreatic beta cells (*Illies et al., 2007*). In another instance deletion of IP6K2, but not the other two IP6K isozymes, reduced cell death (*Nagata et al., 2005*). Whether these observations are due to tissue-specific expression levels, localized cellular pools of PP-InsPs, or distinct modes of action by which the individual IP6Ks operate often remains elusive.

Although genetic experiments perturbing IP6K function have contributed significantly to our understanding of PP-InsP signaling, these studies can be accompanied by compensatory effects and secondary genetic changes. The undesired transcriptional up- and downregulation of numerous genes has caused confounding results in the past (*Worley et al., 2013*). Furthermore, genetic knockout inevitably abolishes the non-catalytic functions of the kinase, which, under some circumstances, are essential for the regulation of specific cellular processes (*Koldobskiy et al., 2010*; *Luo et al., 2001*; *Zhu et al., 2016*; *Chakraborty et al., 2008*; *Fu et al., 2015*). The generation of single point kinase-dead mutants usually warrants correct protein expression and folding, but can be accompanied by notable transcriptional alterations (*Worley et al., 2013*). Lastly, a limitation of genetic approaches is the long timeframe, which can lead to genetic compensation (*El-Brolosy and Stainier, 2017*) and appears inherently incompatible with the rapid interconversion of PP-InsPs in cells, indicating fast signal transmission (*Menniti et al., 1993*; *Glennon and Shears, 1993*).

A well-suited approach to address these challenges is the pharmacological inhibition of IP6K activity. While the benchmark inhibitor *N*2-(*m*-trifluorobenzyl)-*N*6-(*p*-nitrobenzyl)purine (TNP; *Padmanabhan et al., 2009*; *Figure 1b*) has lost appreciation due to the accumulation of unfavorable properties (*Zhu et al., 2017*; *Chang et al., 2002*; *Stork and Li, 2010*), the development of novel pharmacological tools targeting IP6Ks has recently picked up speed (*Kröber et al., 2022*). High-throughput screens and structure-guided design have yielded compounds with low nanomolar potencies, such as SC-919 (*Moritoh et al., 2021*; *Terao et al., 2018*; *Figure 1b*) and others (*Zhou et al., 2022*; *Liao et al., 2021*). However, with the exception of one isozyme-selective inhibitor targeting IP6K1 (Barrow-24; *Wormald et al., 2019*; *Figure 1b*), all other IP6K inhibitors lack specificity.

To achieve selective kinase inhibition, Shokat and co-workers developed the analog-sensitive approach, which enables the generation of mutually selective kinase-inhibitor pairs (*Bishop et al., 1998*; *Lopez et al., 2014*). A conserved medium- to large-sized gatekeeper residue is mutated to glycine or alanine, thereby creating an additional pocket in the ATP-binding site that does not occur in any other wild-type (WT) kinase. While designed inhibitor analogs with a sterically demanding

substituent fit into the enlarged ATP-binding site, they clash with the active site of WT kinases (*Bishop et al., 1998*; *Bishop et al., 1999*; *Liu et al., 1999*). Implementation of this method for various kinase families of multiple organisms has provided remarkable insight into the function and regulation of phosphorylation-based signaling (*Liu et al., 2004*; *Sreenivasan et al., 2003*; *Carroll et al., 2001*; *Weiss et al., 2000*).

While the analog-sensitive approach has been applied to over 80 protein kinases, the implementation for kinases phosphorylating small molecules, such as lipids or sugars, proved more challenging and has received scant attention to date. For example, mutation of the gatekeeper residue in *S. cerevisiae* VPS34, a PI3 kinase, yielded an active mutant allele, however, mutation of the mammalian PI3K p110$\alpha$ gatekeeper resulted in a drastic decline of kinase activity (*Alaimo et al., 2005*). A similar observation was made with PI3K-like kinase TOR2 from *S. cerevisiae*, which was sensitized to the mTOR inhibitor BEZ235 by mutating its gatekeeper residue to alanine. The mammalian kinase mTOR, on the other hand, was rendered catalytically inactive by the same mutation (*Kliegman et al., 2013*).

Here, we implemented the analog-sensitive approach for IP6Ks to identify isozyme-selective inhibitors. While conventional alanine and glycine gatekeeper mutations resulted in a loss of kinase activity, a leucine to valine mutant remained active. A high-throughput screen with over 50,000 compounds yielded a potent and mutant-selective inhibitor of unknown biological activity. Validation and biochemical characterization revealed an allosteric mode of action, which was corroborated by hydrogen deuterium exchange mass spectrometry (HDX-MS) measurements. This non-competitive mechanism of inhibition constitutes a promising new opportunity to selectively target the mammalian IP6K isozymes.

## Results

### Conventional gatekeeper mutations render IP6Ks catalytically inactive

The gatekeeper identification in inositol phosphate (InsP) kinases is facilitated by the structural similarity of their active site architecture to protein kinases (*Randall et al., 2020*). While the exact structures of mammalian IP6Ks have remained elusive to date, those of other members of the InsP kinase family, for example *Homo sapiens* IPMK and *Entamoeba histolytica* IP6KA, have successfully been deciphered (*Ciobanasu et al., 2018*; *Wang et al., 2014*; *Blind, 2020*; *Seacrist and Blind, 2018*). Owing to the decent sequence similarity within the active site, the gatekeeper residues of the human IP6K isozymes could be identified by simple sequence alignment with the abovementioned related InsP kinases and are Leu210, Leu206, and Leu201 for IP6K1, IP6K2, and IP6K3, respectively (*Figure 2a*).

The gatekeeper mutants L210G and L210A of IP6K1 were generated by site-directed mutagenesis and the enzymes were recombinantly expressed, analogous to the WT kinase, as MBP-fusion proteins. To assess their catalytic activity, we used an NMR assay that harnesses fully $^{13}$C-labeled InsP$_6$ as a substrate and enables direct product readout of 5PP-InsP$_5$ (*Figure 2b* and *Figure 2—figure supplement 1*; *Puschmann et al., 2019*; *Harmel et al., 2019*). Unfortunately, both the alanine and glycine mutations were not tolerated by IP6K1 and rendered the protein virtually inactive (*Figure 2c*). Likewise, IP6K2 suffered a drastic decline in substrate turnover upon gatekeeper mutation to glycine (*Figure 2—figure supplement 2*). While the cysteine mutant amenable for the electrophile-sensitive approach (*Garske et al., 2011*) retained residual catalytic activity, established electrophile-sensitive kinase inhibitors did not inhibit the mutant kinase (*Figure 2—figure supplement 3*).

Since the size of the gatekeeper appeared to be relevant, the residue was subsequently mutated to valine; the last remaining hydrophobic amino acid larger than alanine but smaller than leucine. Although the change is comparably subtle, it could still have an impact on inhibitor selectivity. Unlike the previous mutations, the valine gatekeeper mutant completely retained its catalytic activity compared to the WT enzyme and was thus further investigated (*Figure 2c*).

### IP6K1$^{L210V}$ is not susceptible to established analog-sensitive kinase inhibitors

To assess the impact of the valine gatekeeper mutation on the affinity of IP6K1 for ATP, Michaelis–Menten kinetics were measured for IP6K1$^{L210V}$ and IP6K1$^{wt}$ (*Figure 2d*). The $K_M$ value of IP6K1$^{wt}$ was determined as 186 ± 23 µM and is in good accordance with the literature (*Harmel et al., 2019*; *Wormald et al., 2017*; *Wundenberg et al., 2014*). The mutation of the gatekeeper residue did indeed

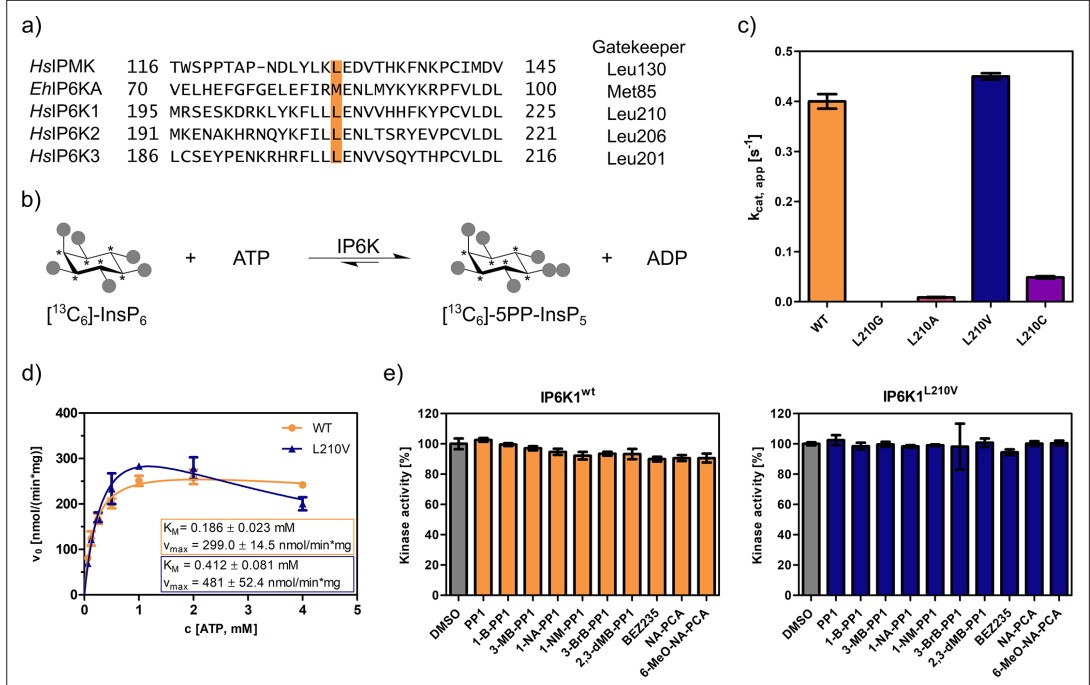

**Figure 2.** An unusual valine gatekeeper mutant retains catalytic activity. (**a**) Sequence alignment of InsP kinases to identify the gatekeeper position of human inositol hexakisphosphate kinases (IP6Ks). The gatekeeper residue is highlighted in orange. (**b**) Kinase reaction of IP6Ks using uniformly $^{13}$C-labeled InsP$_6$ as substrate. Asterisks indicate $^{13}$C-labeled positions. The conversion to 5PP-InsP$_5$ was followed by an established spin-echo difference NMR method (**Harmel et al., 2019**; **Riemer et al., 2021**). (**c**) Catalytic activity of IP6K gatekeeper mutants indicated by apparent turnover numbers $k_{cat, app}$. (**d**) Michaelis–Menten graphs for IP6K1$^{wt}$ and IP6K1$^{L210V}$. (**e**) Screening of established analog-sensitive kinase inhibitors (**Figure 2—figure supplement 4**) at 10 µM concentration against IP6K1$^{wt}$ and IP6K1$^{L210V}$ using the NMR assay. All data points were measured in independent triplicates and error bars represent the standard deviation.

The online version of this article includes the following figure supplement(s) for figure 2:

**Figure supplement 1.** Kinase reaction of inositol hexakisphosphate kinases (IP6Ks) using fully $^{13}$C-labeled InsP$_6$ as a substrate.

**Figure supplement 2.** Catalytic activities of IP6K2 wild-type (WT) and gatekeeper mutants as indicated by the apparent turnover number.

**Figure supplement 3.** Screening of established electrophile-sensitive kinase inhibitors at 10 µM concentration against IP6K1$^{L210C}$ using the NMR assay.

**Figure supplement 4.** Chemical structures of established analog-sensitive kinase inhibitors screened against IP6K1$^{L210V}$.

---

impact the behavior of IP6K1 toward ATP, although the effects were relatively minor. We observed a twofold reduction in ATP affinity ($K_M = 412 \pm 81$ µM) along with a 60% increase in $v_{max}$ from 300 to 480 nmol min$^{-1}$ mg$^{-1}$, implicating a faster substrate conversion despite impaired ATP binding. The valine mutation also seems to exacerbate the substrate inhibition, which complicates the comparison of these steady-state parameters and suggests a change in protein dynamics upon gatekeeper mutation. Compared to many protein kinases, however, which often experienced a substantial reduction in ATP affinity upon gatekeeper mutation in the past (**Bishop et al., 2000**), the change of kinetic parameters between IP6K1$^{wt}$ and IP6K1$^{L210V}$ can be considered minor.

In a preliminary inhibitor screen, an array of established analog-sensitive kinase inhibitors was tested against both IP6K1$^{wt}$ and IP6K1$^{L210V}$. Among them were the most frequently used pyrazolopyrimidine (PP) analogs, BEZ235 and two 5-aminopyrazolo-4-carboxamide inhibitors (**Figure 2—figure supplement 4**). As expected, none of these compounds exhibited any inhibitory activity against IP6K1$^{wt}$ at 10 µM concentration (**Figure 2e**). However, inhibition of IP6K1$^{L210V}$ was not observed either, making these molecules unsuitable as inhibitors for analog-sensitive IP6Ks. Consequently, a novel

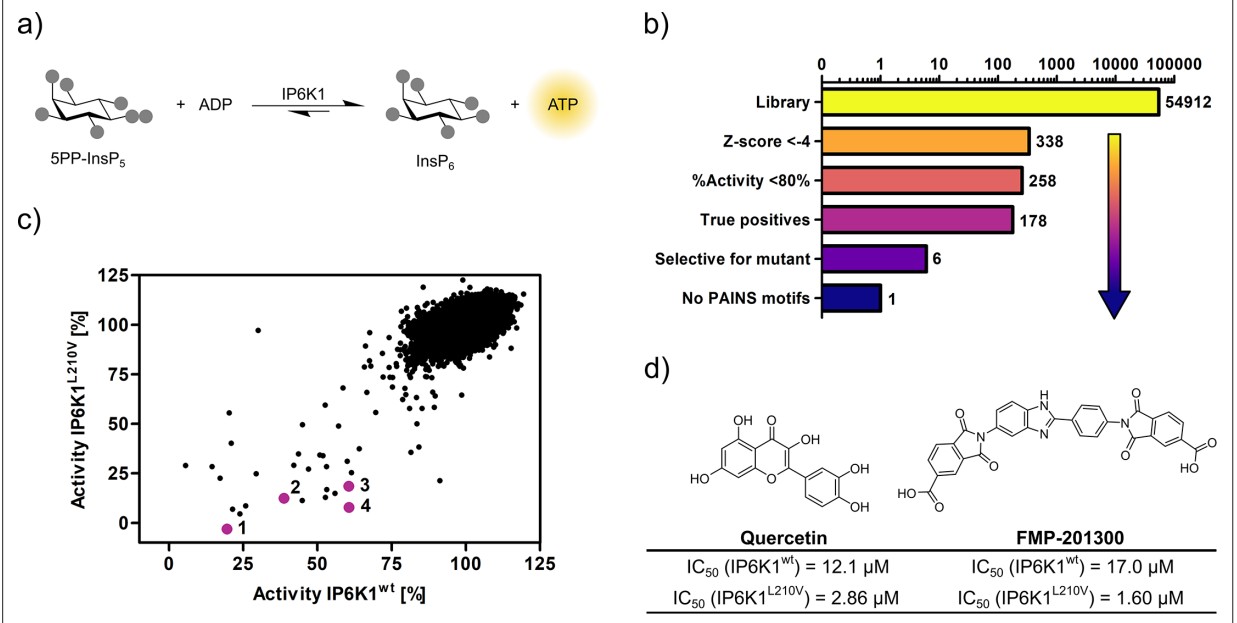

**Figure 3.** High-throughput screening provides gatekeeper mutant-selective hit compound. (**a**) ATP synthase reaction used for the high-throughput screen. The generation of ATP was monitored using a luminescence-based Kinase-Glo assay. (**b**) Reduction of putative hits by applying specific thresholds and manual selection. (**c**) Scatter plot comparing the potency of compounds measured in the primary screen against IP6K1$^{wt}$ and IP6K1$^{L210V}$. Hits highlighted in magenta represent known IP6K inhibitors, or promising hits. 1: 6-hydroxy-DL-DOPA, 2: FMP-201300, 3: myricetin, 4: quercetin. (**d**) Examples of true positive hits that display selectivity for IP6K1$^{L210V}$, including known IP6K inhibitor quercetin and newly discovered FMP-201300.

The online version of this article includes the following figure supplement(s) for figure 3:

**Figure supplement 1.** Optimization of assay conditions for high-throughput screening.

**Figure supplement 2.** Z' plate measurement and Bland-Altman plot indicate high assay quality.

inhibitor scaffold that allows for selective inhibition of analog-sensitive IP6Ks over their WT counterparts needed to be identified.

## Luminescence-based assay of reverse kinase reaction enables pilot screen

Seeking to identify an inhibitor selective for the gatekeeper mutant, IP6K1$^{wt}$ and IP6K1$^{L210V}$ were submitted to a high-throughput screen encompassing 54,912 small molecule inhibitors and inhibitor-like structures (**Lisurek et al., 2010**). Although the NMR assay described above provides direct product detection and the ability to distinguish the PP-InsP isomers (**Harmel et al., 2019**), it is inherently slow and thus not compatible with a high-throughput screen. The use of luminescence-based readouts of ATP consumption to measure IP6K activity is error-prone due to the kinase's unusually low affinity for ATP, the sensitivity to the ATP/ADP ratio, and its inherent ATPase activity (**Wundenberg et al., 2014**; **Voglmaier et al., 1996**). Nevertheless, we were able to take advantage of these properties by examining the reverse reaction of IP6K1 with the Kinase-Glo assay, where 5PP-InsP$_5$ and ADP are converted to InsP$_6$ and ATP (**Figure 3a**). Hence, physiologically relevant nucleotide concentrations could be used without causing ambiguity, since the amount of generated ATP was directly dependent on the substrate concentration. Similar $K_M$ values for ATP and ADP should allow for comparability of inhibitor potencies (**Voglmaier et al., 1996**) and a gain-of-signal detection (rather than loss-of-signal) resulted in a much better signal-to-noise contrast in our setting.

After optimizing reaction parameters to ensure a successful and robust high-throughput screen (**Figure 3—figure supplement 1a and b**), an IC$_{50}$ curve of the known IP6K inhibitor myricetin was recorded to determine the accuracy of the reverse reaction (**Figure 3—figure supplement 1d**; **Wormald et al., 2017**; **Gu et al., 2019**). The measured value of 3.1 µM is in the same range as the published value for the forward reaction (**Wormald et al., 2017**), thereby strengthening the initial rationale. To further confirm the assay viability, a Z'-plate was measured employing positive and

negative controls only (*Figure 3—figure supplement 2a*). An excellent Z'-factor (*Zhang et al., 1999*) of 0.81 was achieved indicating high assay quality and feasibility. Before screening the entire library, a pilot screen encompassing 1670 FDA-approved drugs and 1280 pharmacologically active compounds was conducted. It confirmed the suitability of the final screening setup by yielding known IP6K inhibitors myricetin and 6-hydroxy-DL-DOPA (*Wormald et al., 2017*; *Gu et al., 2019*) and displaying good hit reproducibility as depicted in the Bland–Altmann plot (*Figure 3—figure supplement 2b*; *Martin Bland and Altman, 1986*).

## High-throughput screening identifies a mutant-selective inhibitor

The successful pilot screen was followed by screening 54,912 compounds comprising diverse chemical motifs (*Lisurek et al., 2010*) and natural products against IP6K1$^{L210V}$ and IP6K1$^{wt}$. Putative hits for the gatekeeper mutant were subsequently identified by excluding compounds with a z-score higher than −4 and a residual activity greater than 80% (*Figure 3b*). The remaining 258 compounds (0.5% hit rate) were retested in a dose-dependent reconfirmation and counter screen (see SI for details) to eliminate random false positives (compounds that do not confirm activity) and chemical false positives (compounds directly interfering with the assay readout). Additionally, information was added on compounds identified as frequent hitters in our in-house screening data. This analysis uncovered 80 molecules as false positives, while 178 inhibitors were consequently classified true hits. However, only a handful appeared to be selective for IP6K1$^{L210V}$ as evident from the scatter plot comparing the residual activities of both kinases (*Figure 3c*). Curiously, the majority of mutant-selective inhibitors were dietary flavonoids such as quercetin and fisetin (*Supplementary file 1*; *Gu et al., 2019*).

Since flavonoids are promiscuous ATP-competitive inhibitors with various off-target effects (*Xue et al., 2017*; *Navarro-Retamal et al., 2016*; *Wright et al., 2013*; *Russo et al., 2017*; *Walker et al., 2000*), and the compounds had been flagged as pan-assay interference compounds (PAINS) based on structural filters (*Baell and Nissink, 2018*), they were deemed unsuitable as selective IP6K inhibitors and thus not investigated further. The only non-polyphenolic molecule, FMP-201300, displayed good selectivity for the gatekeeper mutant with IC$_{50}$ values of 17 and 1.6 µM against IP6K1$^{wt}$ and IP6K1$^{L210V}$, respectively (*Figure 3d*). Importantly, the almost symmetrical inhibitor has no known inhibitory activities against other biological targets and does not display any recognized PAINS motifs (*Baell and Nissink, 2018*). Furthermore, FMP-201300 was neither redox-active, nor cytotoxic, against HEK293 or HepG2 cell lines at a concentration of 10 µM (*Supplementary file 2*).

## Potency of FMP-201300 is dictated by carboxylic acid functional groups

Next, FMP-201300 was validated with the NMR assay using isotopically labeled [$^{13}$C$_6$]-InsP$_6$ as a substrate, and 1 mM ATP, as well as a creatine kinase/phosphocreatine-based ATP-regenerating system. The assay confirmed FMP-201300 as a potent and mutant-selective inhibitor with IC$_{50}$ values of 114 and 810 nM for IP6K1$^{L210V}$ and IP6K1$^{wt}$, respectively (*Figure 4a*). To evaluate whether the inhibitory behavior of FMP-201300 was transferable to the gatekeeper mutant of another IP6K isozyme, dose response curves against IP6K2$^{wt}$ and IP6K2$^{L206V}$ were recorded. FMP-201300 was 11 time more potent against IP6K2$^{L206V}$ compared to IP6K2$^{wt}$ with IC$_{50}$ values of 190 nM and 2.14 µM, respectively (*Figure 4b*). Due to difficulties in obtaining recombinant catalytically active IP6K3$^{wt}$, this isozyme could not be investigated. However, the evolutionarily related ortholog IP6KA from *Entamoeba histolytica* (*Eh*) was sensitized to inhibition by FMP-201300 when its methionine gatekeeper was mutated to valine (*Figure 4—figure supplement 1a*). The approach therefore appears transferable not only between the mammalian isozymes, but also to other IP6K orthologs.

Given these promising initial results, we wanted to understand which elements of the peculiar structure of FMP-201300 are potentially involved in steering inhibitory activity toward the gatekeeper mutant. Therefore, the MolPort chemical library was screened for compounds containing a structure similar to FMP-201300 or core substructures such as phthalimides or benzimidazoles. Four analogs (**A1–A4**) were subsequently tested against IP6K1$^{L210V}$ at a single concentration of 10 µM (*Figure 4c*). Interestingly, none of the analogs inhibited the kinase by more than 30%. Although **A1** was merely lacking the carboxylic acid groups, it was completely inactive. The only analog still possessing a carboxylic acid moiety, **A2**, inhibited IP6K1$^{L210V}$, albeit with severely reduced potency. This result is consistent with the observation made with IP6K inhibitors SC-919 (*Terao et al., 2018*) and LI-2172

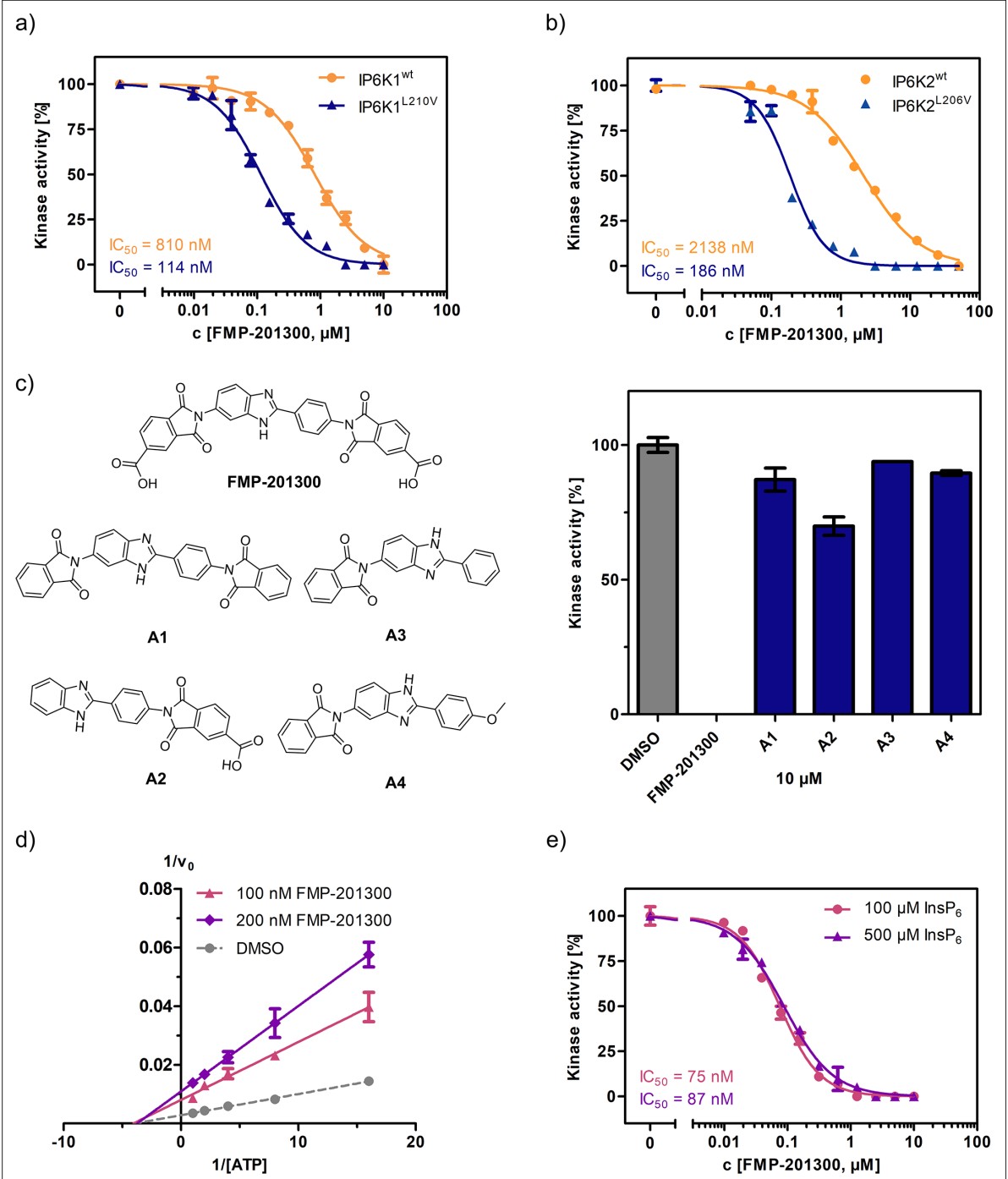

**Figure 4.** FMP-201300 appears to have very specific interactions with the kinase. (**a**) IC$_{50}$ curves of FMP-201300 against IP6K1$^{wt}$ and IP6K1$^{L210V}$. (**b**) IC$_{50}$ curves of FMP-201300 against IP6K2$^{wt}$ and IP6K2$^{L210V}$. 100% activity corresponds to the DMSO control and 0% indicates no substrate conversion. (**c**) Structures of FMP-201300 analogs and their inhibitory activities against IP6K1$^{L210V}$ at 10 μM concentration. (**d**) Lineweaver–Burk plot of FMP-201300 against IP6K1$^{L210V}$ at two different inhibitor concentrations. The plotted lines for the DMSO control and two different inhibitor concentrations intersect almost precisely on the x-axis, indicating no change in $K_M$ value and a decrease in $v_{max}$ upon inhibition (**e**) IC$_{50}$ curves of FMP-201300 against IP6K1$^{L210V}$ at two different InsP$_6$ concentrations. All data points were measured in independent triplicates and error bars represent the standard deviation.

The online version of this article includes the following source data and figure supplement(s) for figure 4:

**Figure supplement 1.** Characterization of the interaction between FMP-201300 and *Eh*IP6KA.

**Figure supplement 1—source data 1.** Data collection and structure refinement statistics of *Eh*IP6KA$^{M85V}$ crystallization.

*Figure 4 continued on next page*

*Figure 4 continued*

**Figure supplement 2.** Established pan-IP6K inhibitor TNP displays ATP-competitive mechanism and no selectivity towards the valine gatekeeper mutant.

**Figure supplement 3.** Reverse transcription quantitative PCR analysis to measure 45S pre-rRNA transcript levels using two different primer sets.

(*Liao et al., 2021*), which share the benzoic acid substructure with FMP-201300. Hence, the carboxylic acid appears to be indispensable for inhibitory activity and is likely engaged in pivotal interactions at the inhibitor-binding site.

## Kinetic characterization reveals an allosteric mode of inhibition

Before undertaking further analyses to elucidate the structural determinants of mutant selectivity, we wanted to confirm the ATP-competitive inhibition mechanism of FMP-201300. Therefore, Michaelis–Menten measurements in the presence of inhibitor were performed and plotted reciprocally to obtain Lineweaver–Burk graphs (*Lineweaver and Burk, 1934*). Surprisingly, these measurements revealed a clear allosteric mechanism for FMP-201300 (*Figure 4d*). To dispel any doubts about the accuracy of the Lineweaver–Burk plots, the assay was performed on the established IP6K inhibitor TNP. In agreement with the literature, the results indicated an unambiguous competitive mechanism of action for TNP (*Figure 4—figure supplement 2a*). In addition, TNP exhibited no significant difference in potency against IP6K1$^{wt}$ and IP6K1$^{L210V}$, again pointing toward a different inhibitory mechanism of FMP-201300 (*Figure 4—figure supplement 2b*).

These results suggest that the leucine to valine gatekeeper mutation – despite being buried deep in the ATP-binding pocket – must have caused a conformational change within the kinase structure that facilitates binding of FMP-201300. The fact that the inhibitor exhibits the same behavior against IP6K2 and *Eh*IP6KA diminishes the likelihood of a distant allosteric site, since IP6K1 and IP6K2 only share 47% sequence identity, most of which lies within the catalytic region. Since a structural alteration at a vastly remote site of IP6K1, caused by the subtle amino acid change, appears unlikely, we hypothesize that the compound binds adjacent to the ATP-binding site.

Although FMP-201300 does not resemble the densely charged InsP$_6$ molecule, the two carboxylic acids could potentially contribute to a substrate-competitive mechanism. Due to the nanomolar $K_M$ of IP6K1 for InsP$_6$ (*Voglmaier et al., 1996*) and the inherently low sensitivity of the NMR assay, Michaelis–Menten measurements were not feasible. The Cheng–Prusoff equation states that for [S] >> $K_M$, the IC$_{50}$ values correlate linearly with substrate concentration for competitive inhibitors (*Cheng and Prusoff, 1973*). However, the difference in IC$_{50}$ values measured at 100 and 500 µM InsP$_6$ concentration was negligible, excluding substrate inhibition as a potential mechanism (*Figure 4e*).

## HDX-MS corroborates allosteric mechanism

For a more detailed picture of the specific binding modalities of FMP-201300 and the factors promoting selectivity for the gatekeeper mutant, co-crystal structures of inhibitor bound to the kinase would undoubtedly be beneficial. However, to date the mammalian IP6K isozymes have been recalcitrant in crystallization efforts. This limitation may, in part, be circumvented by focusing on *Eh*IP6KA, which has been crystallized and structurally analyzed (*Wang et al., 2014*). An alignment of the AlphaFold structure model of IP6K1 and the crystal structure of *Eh*IP6KA suggests significant structural conservation, especially within the active site (*Figure 4—figure supplement 1c*). The fact that FMP-201300 recapitulates its gatekeeper mutant selectivity and allosteric binding mode against *Eh*IP6KA$^{M85V}$ (*Figure 4—figure supplement 1a, b*) prompted us to attempt co-crystallization of this related kinase and the inhibitor. While the valine gatekeeper mutant of *Eh*IP6KA was readily crystallized using the published conditions for the WT counterpart (*Wang et al., 2014*), soaking or co-crystallization with FMP-201300 remained unsuccessful. Remarkably, the structures of the WT and mutant *Eh*IP6KA are virtually identical except for the additional space generated by the reduced gatekeeper size (*Figure 4—figure supplement 1d*). This result illustrates a potential limitation of crystallography, as it can only provide a snapshot of a momentary protein conformation and may not be suited to unveil subtle dynamic changes that occur upon inhibitor binding.

We therefore utilized HDX-MS to elucidate which parts of the protein participate in inhibitor binding. Initially, we compared the deuterium exchange rates between the IP6K1$^{L210V}$ apo and IP6K1$^{L210V}$ bound to FMP-201300. Major decreases in exchange rates were observed in proximity to

the ATP-binding site, whereas remote sites were mostly unaltered (*Figure 5a* and *Figure 5—figure supplement 1a*). Amino acid residues 40–45, 67–79, and 205–212, corresponding to β strands 1–3, and residues 53–58 within the $\alpha$C helix of IP6K1, exhibited large decreases in deuterium exchange rates. Importantly, this includes the gatekeeper residue Leu210 and parts of the hinge region (Glu211–Asn212–Val213), corroborating the hypothesis of ATP-adjacent binding. In agreement with previous kinetic measurements that suggested a non-competitive mechanism, changes in ATP-binding regions were less pronounced with no differences that met the criteria for significance for the ATP ribose-binding residue Asp224 (*Figure 5—figure supplement 1*).

Decreases in HD exchange could be indicative of direct interactions of the inhibitor with IP6K1, which would support an allosteric mode of action. Specifically, FMP-201300 might bind in a cleft next to the ATP site, which stretches to β1–β3 strands and $\alpha$C helix, while engaging with the gate-keeper residue. Similar binding modes have been observed for other allosteric regulators, including epidermal growth factor receptor (EGFR) and mitogen-activated protein kinase (MEK) inhibitors (*Jia et al., 2016*; *Rice et al., 2012*). Interestingly, crucial structural features of IP6K1 such as the IP helices and the PDKG catalytic motif did not exhibit differences in deuterium exchange rates, indicating no direct binding interaction or induced structural changes at these sites (*Figure 5—figure supplement 1*). Conversely, the conserved IDF tripeptide (Ile398–Asp399–Phe400) was moderately affected by binding of FMP-201300, potentially influencing ATP binding as Asp399 coordinates two $Mg^{2+}$ ions and makes polar contacts with the nucleotide phosphates. Furthermore, a unique $3_{10}$A helix comprising residues 278–284 showed a medium difference in deuterium exchange rates. This helix is located opposite the IP helices in the substrate-binding pocket and makes pivotal interactions with $InsP_6$ (*Figure 5—figure supplement 1*). Since previous biochemical experiments excluded substrate competition of FMP-201300, this region likely undergoes dynamic structural changes rather than being involved in direct binding. Nevertheless, these structural changes could conceivably affect the kinetic properties of IP6K1, as residues Lys278 and Arg282 are involved in the positioning of $InsP_6$. Hence, a change in their orientation could impact $InsP_6$ binding and thereby kinase catalytic activity.

To understand the impact of the gatekeeper mutation on protein conformation, we next investigated the differences in deuterium exchange rates between the IP6K1$^{wt}$ and IP6K1$^{L210V}$ apo forms. The leucine to valine mutation led to increases in HD exchange in the β2–β3 strands and $\alpha$C helix, indicating an overall increased flexibility in this region (*Figure 5*). A minor decrease in exchange was also identified at a flexible region immediately adjacent to the $\alpha$C helix (residues 59–68) that flanks the active site and is a dynamic modulator of kinase activity (*Taylor and Kornev, 2011*; *Kornev et al., 2006*). Despite the minor differences between the structures of the gatekeeper residues, the leucine appears to be important for the stability of these features, although it is difficult to model without a high-resolution structure of the human enzyme. The associated structural rearrangements induced by the gatekeeper mutation could be responsible for the mutant selectivity of FMP-201300, since targeting the displacement of $\alpha$C helices has been a viable approach for the development of several allosteric inhibitors (*Palmieri and Rastelli, 2013*). For instance, the T790M resistance gatekeeper mutation in EGFR sensitized the tyrosine kinase to allosteric inhibitors by an outward displacement of the $\alpha$C helix in the inactive conformation (*Jia et al., 2016*). Small increases in deuterium uptake also occurred in a flexible loop adjacent to the $\alpha$C helix and a distant helix that could be important for substrate binding (residues 321–330 and 277–285), indicating further increased flexibility of the secondary structure in the mutant enzyme that allows for the accommodation of FMP-201300, ATP, and substrate (*Figure 5*).

Finally, we wondered whether FMP-201300 would impose similar changes in deuterium uptake on IP6K1$^{wt}$. We therefore compared the apo kinase to the inhibitor-bound form. With the exception of the β2 and β3 strands and parts of the hinge region that displayed a slight decrease in HDX, no significant changes in deuterium incorporation were observed (*Figure 5—figure supplement 1a*). These data underline the selectivity and preferred binding of FMP-201300 to IP6K1$^{L210V}$. In conclusion, the L210V gatekeeper mutation of IP6Ks putatively causes a displacement of the $\alpha$C helix sufficient to expand a pocket adjacent to the ATP-binding site, which sensitizes the kinase to an allosteric inhibitor. Alternatively, the mutation induces a certain degree of flexibility that facilitates the induced-fit binding and $\alpha$C helix displacement by FMP-201300.

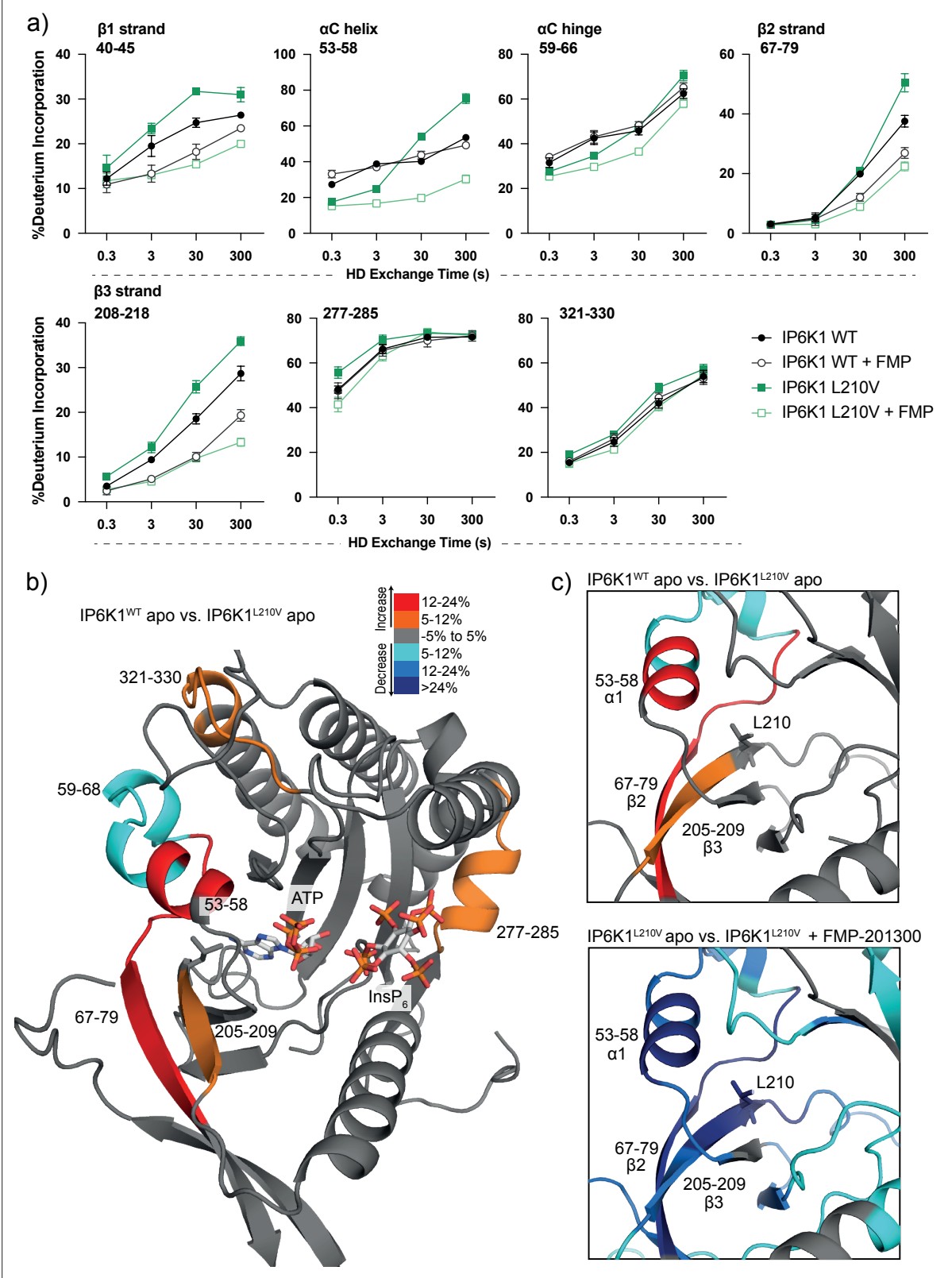

**Figure 5.** Hydrogen deuterium exchange mass spectrometry (HDX-MS) reveals increased flexibility in the β2–β3 strands and αC helix induced by the IP6K1[L210V] mutation that sensitize the enzyme to FMP-201300. (**a**) HDX differences in IP6K1. Time course of deuterium incorporation for a selection of peptides. Raw data can be found in the **Supplementary file 1**. (**b**) Overall HDX-MS changes in deuterium incorporation induced by the IP6K1[L210V] mutation. Differences in deuterium exchange rates mapped on a model of IP6K1[wt] (AlphaFold structure prediction Q92551 with ATP and InsP$_6$ from

*Figure 5 continued on next page*

Figure 5 continued

*Eh*IP6KA docked in the active site (PDB: 4O4F)). Peptides that showed significant differences in HDX and met the cut-offs were included (>6% deuterium incorporation and 0.5 Da with an unpaired Student's *t*-test of p < 0.05). Strongly disordered regions and regions of low per-residue confidence scores (pLDDT) were omitted for clarity. (**c**) A magnified view of gatekeeper region included the β2–β3 strands and αC helix. The same regions that show increased flexibility in the IP6K1[L210V] mutant (top panel) also undergo large decreases in exchange upon binding to FMP-201300 (bottom panel). This increased flexibility likely allows accommodation of the inhibitor and ATP.

The online version of this article includes the following source data and figure supplement(s) for figure 5:

**Source data 1.** Hydrogen deuterium exchange mass spectrometry raw data.

**Figure supplement 1.** Hydrogen deuterium exchange mass spectrometry (HDX-MS) shows FMP-201300 binding to IP6K1[L210V] leads to decreases that correspond to an allosteric mechanism of action.

## Discussion

We have investigated the suitability of an analog-sensitive approach for the selective inhibition of the mammalian IP6K isozymes. While the conventional glycine and alanine gatekeeper mutants suffered a drastic decline in catalytic efficiency, a subtle leucine to valine mutation did not perturb the kinase activity. After established analog-sensitive kinase inhibitors failed to target IP6K1[L210V], a high-throughput screen uncovered FMP-201300 as a potent and mutant-selective inhibitor. The compound had not been assigned any function in previous screens and displayed no toxicity in cell culture. A preliminary structure–activity analysis indicated the indispensability of the carboxylic acid as a structural motif required for inhibitory potency. This phenomenon had been observed in previous studies investigating IP6K inhibitors, where the removal of a carboxylate moiety from the core structure resulted in a drastic decrease in potency of up to 2600-fold (*Zhou et al., 2022*; *Liao et al., 2021*). While this functional group appears essential for inhibitory activity, it can pose several drawbacks such as metabolic instability, toxicity, and poor passive diffusion across biological membranes. To assess cell permeability and cellular activity in a preliminary experiment, we measured the reduction of ribosomal RNA synthesis upon treatment with FMP-201300 (*Figure 4—figure supplement 3*). The inhibitor, alongside two established IP6K inhibitors, was able to recapitulate this recently reported phenotype (*Morgan et al., 2022*), indicating sufficient crossing of the cell membrane despite its two carboxylic acids. In fact, other carboxylate-containing IP6K inhibitors had proven efficacious *in cellula* and in vivo before (*Moritoh et al., 2021*; *Zhou et al., 2022*). Regardless, a wide range of bioisosteres, including hydroxamic acids, sulfonamides, tetrazoles, and others (*Ballatore et al., 2013*), would be available to circumvent potential issues arising with FMP-201300.

Unexpectedly, Lineweaver–Burk measurements suggested an allosteric mechanism of FMP-201300, which appeared counterintuitive with regard to the analog-sensitive approach. After an InsP$_6$-competitive mode of action had been ruled out, attempts to gain structural insights by crystallographic analysis of the ortholog *Eh*IP6KA were undertaken. Although the valine gatekeeper mutant of *Eh*IP6KA was successfully crystallized, soaking, or co-crystallization of FMP-201300 could not be achieved. Lastly, HDX-MS measurements were performed to obtain information on the structural determinants of mutant selectivity. This analysis revealed distinct interaction profiles of FMP-201300 with the WT and mutant kinase and suggested an αC helix displacement to be responsible for sensitizing IP6K1 to allosteric inhibition.

The reduction of catalytic activity upon introduction of a space-creating gatekeeper mutation is by no means an issue exclusively occurring in IP6Ks. In fact, a loss in activity was observed for roughly 30% of all examined protein kinases (*Zhang et al., 2005*) and has proven particularly detrimental for small molecule kinases, such as the phosphoinositide lipid kinases (*Alaimo et al., 2005*; *Kliegman et al., 2013*). To address this challenge, Shokat and co-workers identified second-site suppressor mutations that rescue the catalytic activity of some kinases sensitive to the reduction in gatekeeper size (*Zhang et al., 2005*). Since IP6Ks possess a protein kinase fold (*Randall et al., 2020*), they might be amenable to second-site suppressor mutations as well. As an alternative, we implemented a valine gatekeeper mutation that was well tolerated by IP6Ks. This unconventional approach could, in fact, constitute a viable alternative for kinases that are incompatible with the classical gatekeeper mutations. While the substitution of leucine with valine was comparatively subtle, the majority of kinases in the human kinome carry medium- to large-sized gatekeeper residues, such as methionine, tyrosine, or phenylalanine (*Zuccotto et al., 2010*).

Although the analog-sensitive method is a useful tool in itself, the prospect of having FMP-201300, or analogs thereof, as well-defined allosteric inhibitors is also very promising. The greater heterogeneity of residues and conformations in allosteric pockets (*Lu et al., 2014*), even in isozymes of the same kinase, could pave the way for isozyme-selective IP6K inhibitors that do not require chemical genetic engineering. However, the kinetic measurements and HDX-MS analysis are insufficient to confirm the exact binding position with certainty, necessitating further biophysical measurements such as limited proteolysis MS (*Malinovska et al., 2023*; *Schopper et al., 2017*), cryo-EM (*Glaeser, 2016*; *Bai et al., 2015*), or NMR using $^{15}$N-labeled protein.

Binding of FMP-201300 to a hydrophobic pocket adjacent to the ATP site, which is generated by $\alpha$C helix displacement, seems plausible and is an established mode of action for allosteric kinase inhibitors (*Palmieri and Rastelli, 2013*). It remains unclear if the gatekeeper mutation caused this displacement or merely increased the conformational flexibility of the $\alpha$C helix to facilitate the induced fit of FMP-201300. The first scenario appears more likely, as a similar phenomenon was observed for an allosteric EGFR inhibitor that was 1000-fold more potent against the acquired T790M gatekeeper resistance mutant, compared to the WT kinase (*Jia et al., 2016*; *Zhao et al., 2021*). The EGFR inhibitor also exhibited direct binding interaction with the mutated gatekeeper residue, similar to our observations with FMP-201300. In general, these allosteric inhibitors preferably target an inactive conformation of protein kinases, which often confers selectivity owing to their greater structural heterogeneity compared to the corresponding active conformations (*Laufkötter et al., 2022*). It could therefore be worthwhile to investigate and allosterically target the inactive conformation of IP6Ks.

The biochemical characterization of FMP-201300 has taken a surprising yet fascinating turn that necessitates follow-up investigations. It needs to be assessed whether the structural change putatively caused by the gatekeeper mutations has an impact on known protein–protein interactions and scaffolding functions of IP6Ks. Furthermore, it will be interesting to see if this allosteric pocket can be efficiently targeted without prior mutation of the gatekeeper. Lastly, a suitable cellular phenotype to study the inhibitor's efficacy and specificity in a biological setting would be highly desirable. As of now, FMP-201300 constitutes a valuable addition to the ever-growing group of IP6K inhibitors and serves as an intriguing starting point for further characterization and improvement. It could provide a springboard for the heavily sought-after isozyme-selective inhibition of mammalian IP6Ks, either as analog-sensitive kinase inhibitor or allosteric lead compound for the development of inhibitors against the WT kinases.

## Materials and methods

All reagents were purchased from common suppliers such as Sigma-Aldrich, VWR, Roth, TCI, Thermo Scientific, or Roche, and used without further purification unless stated otherwise. The entire PCR and LIC equipment (5× Phusion GC buffer, ThermoPol buffer, dNTP's, Phusion DNA polymerase, Taq DNA polymerase, SspI, T4 DNA polymerase) and Dpn1, CutSmart Buffer, and SOC medium were purchased from New England Biolabs (NEB). All PP1 inhibitor analogs were acquired from Toronto Research Chemicals and BEZ235 from Cayman Chemical. Deuterated solvents were purchased from Deutero. Luminescence-based detections were performed with Kinase-Glo assays from Promega.

The pET-His6-MBP-N10-TEV LIC cloning vector (2C-T) was a gift from Scott Gradia (Addgene plasmid # 29706; http://n2t.net/addgene:29706; RRID:Addgene_29706).

PCR was performed on a Bio-Rad C1000 Touch Thermal Cycler. Concentrations of DNA and proteins were determined spectroscopically with a Thermo Scientific NANODROP 2000C Spectrophotometer. All recombinant proteins were purified on a Bio-Rad NGC chromatography system with a BioFrac Fraction Collector. NMR spectra were recorded on a Bruker SUPERSHIELD 600 PLUS. Luminescence assays were read out with a Tecan Infinite M Plex reader. Diffraction data were collected at 100 K at the beamline BL14.1 operated by the Helmholtz-Zentrum Berlin (HZB) in the BESSY II electron storage ring (Berlin-Adlershof, Germany) (*Mueller et al., 2012*), using a wavelength of 0.9184 Å. Mass spectrometry experiments were performed on a Thermo Scientific Orbitrap Elite.

### Ligation-independent cloning

The LIC cloning vectors were linearized by SspI digest following the manufacturers protocol. After incubation for 30 min at 37°C, the linearized plasmids were purified with the QIAquick PCR Purification Kit

(50). The IP6K open reading frames (ORFs) were amplified by PCR from the pTrc-His constructs using Phusion DNA polymerase and the primers listed in *Supplementary file 3*. After initialization at 98°C for 1 min, the PCR cycle consisted of denaturation for 10 s at 98°C, annealing for 20 s at 65°C, and elongation for 45 s at 72°C, with 30 repetitions. Final elongation was performed for 5 min at 72°C. The amplified inserts were purified by PCR purification, DpnI digest, and another PCR purification. The complementary overhangs were created by T4 DNA polymerase digest adding dGTP to the linearized plasmid reaction and dCTP to the insert reaction. After incubation for 30 min at 22°C, T4 DNA polymerase was heat inactivated by incubation for 20 min at 75°C. Linearized plasmid and IP6K ORF were ligated by combing them in a molar ratio of 1:2 (vector:insert) and incubating them for 30 min at room temperature. After addition of 1 μl ethylenediaminetetraacetic acid (EDTA, 25 mM), 5 μl were used for transformation into *E. coli* TOP10. The successful subcloning was verified by colony PCR from the resulting colonies using Taq DNA polymerase with ThermoPol buffer and the LIC primers listed in *Supplementary file 3*. After 8 min initialization at 98°C, the PCR cycle consisted of denaturation for 30 s at 95°C, annealing for 30 s at 65°C, and elongation for 90 s at 68°C with 30 repetitions. Final elongation was performed for 5 min at 68°C.

## PCR mutagenesis

Plasmid DNA harboring the corresponding IP6K ORF was extracted from an overnight culture of the pET-His$_6$-MBP-N$_{10}$-TEV (*Hs*IP6K) or pET15b-His$_6$ (IP6KA) vector in TB-Amp using a *QIAGEN* Miniprep kit. Single point mutations were installed by employing the primers listed in *Supplementary file 3*. 50 μl PCR reactions were performed on a *Bio-Rad* C1000 Touch Thermal Cycler following the *NEB* Phusion High-Fidelity DNA polymerase protocol using the following temperature program: 5 min 98°C → 30 s 98°C → 3 min 72°C → Cycle to step 2 30× → 5 min 72°C.

## Expression and purification of MBP-IP6K constructs

A 25-ml overnight culture of *E. coli* Arctic Express harboring the IP6K construct on a pET-His$_6$-MBP-N$_{10}$-TEV vector in TB-Amp/Gen was diluted to a final OD$_{600}$ of 0.05 and grown to OD$_{600}$ of 0.7 at 37°C. The temperature was switched to 13°C and expression induced with 0.2 mM isopropyl β-D-1-thiogalactopyranoside (IPTG) after 1 hr. After 20 hr expression, the cells were pelleted by centrifugation (3000 × *g*, 10 min, 4°C), washed with cold lysis buffer (20 mM Tris–HCl pH 7.4, 150 mM NaCl) and centrifuged again. The pellet was resuspended in 10 ml lysis buffer per gram wet weight and supplemented with lysozyme, DNase I and protease inhibitor. After 30 min of incubation on ice, the cells were first homogenized (1× 30 s) and then lysed with a microfluidizerTM LM10 at 15,000 psi with three iterations. The cell debris was removed by centrifugation (20,000 × *g*, 20 min, 4°C) and the supernatant lysate filtered (*VWR* vacuum filter, PES, 0.45 μm). The lysate was loaded onto an equilibrated 5 ml HiTrap IMAC HP column (Ni-NTA, *GE Healthcare*) at a flowrate of 1.5 ml/min. The column was washed with 10 CV wash buffer (20 mM Tris–HCl pH 7.4, 500 mM NaCl, 50 mM imidazole) and the protein eluted with a step gradient of elution buffer (20 mM Tris–HCl pH 7.4, 500 mM NaCl, 500 mM imidazole) in wash buffer with 50% B and 100% B over 5 CV, respectively. Protein-containing fractions were pooled and diluted 20-fold with anion exchange start buffer (20 mM Tris–HCl pH 8.0). The protein was loaded onto an equilibrated 5 ml HiTrap Q FF (*GE Healthcare*) column at a flowrate of 2 ml/min, washed with 3 CV of start buffer and eluted with a 0–100% gradient of elution buffer (20 mM Tris–HCl pH 8.0, 1 M NaCl) in start buffer over 10 CV. Protein-containing fractions were concentrated by spin-filtration through 30 kDa cut-off filters to give 1–4 ml and loaded by injection onto a HiLoad 16/60 Superdex 200 pg column equilibrated with 20 mM Tris–HCl pH 7.4, 500 mM NaCl, 1 mM dithiotreitol (DTT). After elution at a flowrate of 0.5 ml/min, protein-containing fractions were united, concentrated by spin-filtration through 30 kDa cut-off filters, adjusted to 12.5% glycerol, aliquoted and frozen at −80°C. The protein concentration was determined by sodium dodecyl sulfate–polyacrylamide gel electrophoresis with Coomassie staining using a bovine serum albumin (BSA) standard.

## Expression and purification of *Eh*IP6KA constructs

*Eh*IP6KA was expressed as described before (*Harmel et al., 2019*) with slight alterations. An overnight culture of *E. coli* Arctic Express in TB-Amp/Gen harboring the IP6KA construct on a pET15b-His$_6$ vector was diluted 100-fold into 800 ml and grown for 6 hr at 37°C. The temperature was switched to 13°C and expression induced with 0.1 mM IPTG after 1 hr. After overnight expression the cells were

pelleted by centrifugation (3000 × $g$, 10 min, 4°C), resuspended in 10 ml lysis buffer (25 mM Tris–HCl, pH 7.4, 500 mM NaCl, 50 mM imidazole) per gram wet weight and supplemented with lysozyme, DNase I and protease inhibitor. After 30 min of incubation on ice the cells were lysed with a microfluidizer LM10 at 15,000 psi with five iterations. The cell debris was removed by centrifugation (25,000 × $g$, 20 min, 4°C) and the supernatant lysate filtered (*VWR* vacuum filter, PES, 0.45 μm). The clarified lysate was loaded onto an equilibrated 5 ml HiTrap IMAC HP (Ni-NTA, *GE Healthcare*) column at a flowrate of 1 ml/min. The protein was eluted with a gradient of elution buffer (25 mM Tris–HCl pH 7.4, 200 mM NaCl, 500 mM imidazole) in lysis buffer from 0% to 100% over 10 CV. Protein-containing fractions were concentrated by spin-filtration through 3.5 K cut-off filters and dialyzed overnight against dialysis buffer (20 mM Tris–HCl pH 7.4, 200 mM NaCl, 1 mM DTT). The dialyzed protein was adjusted to 25% glycerol, aliquoted and frozen at −80°C. Protein concentration was determined by bicinchoninic acid (BCA) assay using BSA standards for calibration.

## NMR activity assay and $IC_{50}$ measurements for IP6K1 constructs

The NMR assay for IP6K activity was performed similarly as described before (*Harmel et al., 2019*) with slight alterations. Enzyme assays were performed in $D_2O$ in a total volume of 150 μl containing 20 mM 4-(2-hydroxyethyl)-1-piperazineethanesulfonic acid (HEPES) pH* 6.8, 50 mM NaCl, 6 mM $MgCl_2$, 1 mM ATP, 0.2 mg/ml BSA, 1 mM DTT, 5 mM creatine phosphate and 1 U/ml creatine kinase (ATP-regenerating system) and 50 nM kinase if not otherwise stated. HEPES, NaCl, and BSA were prepared as buffer A and DTT and creatine phosphate as buffer B in $D_2O$ and were adjusted to pH* 6.8, respectively. If applicable, inhibitor was added as a DMSO-$d_6$ stock in a twofold dilution series at a final DMSO concentration of 1% if not otherwise mentioned. Otherwise, DMSO-$d_6$ was added to the same percentage. The samples were equilibrated at 37°C for 3 min and the reaction initiated by adding 100 μM [$^{13}C_6$]-InsP$_6$. The reaction was quenched with 400 μl quenching solution (20 mM EDTA pD 6.0, 68.75 mM NaCl) and transferred into an NMR tube before being measured on a *Bruker* SUPERSHIELD 600 PLUS with an NMR pulse program developed previously (*Harmel et al., 2019*) $IC_{50}$ curves and values were calculated in *GraphPad* Prism (Version 5.04.) using nonlinear regression, where the DMSO control defines 0% inhibition and a no kinase control defines 100% inhibition.

## NMR activity assays and $IC_{50}$ measurements for *Eh*IP6KA constructs

The NMR assay for *Eh*IP6KA activity was performed similarly as described before (*Harmel et al., 2019*) with slight alterations. Enzyme assays were performed in $D_2O$ in a total volume of 150 μl containing 20 mM MES pH* 6.4, 50 mM NaCl, 6 mM $MgCl_2$, 1 mM ATP, 0.2 mg/ml BSA, 1 mM DTT, 5 mM creatine phosphate, and 1 U/ml creatine kinase (ATP-regenerating system) and 50 nM kinase if not otherwise stated. MES, NaCl, and BSA were prepared as buffer A and DTT and creatine phosphate as buffer B in $D_2O$ and were adjusted to pH* 6.4, respectively. If applicable, inhibitor was added as a DMSO-$d_6$ stock in a twofold dilution series at a final DMSO concentration of 1% if not otherwise mentioned. Otherwise, DMSO-$d_6$ was added to the same percentage. The samples were equilibrated at 37°C for 3 min and the reaction initiated by adding 100 μM [$^{13}C_6$]-InsP$_6$. The reaction was quenched with 400 μl quenching solution (20 mM EDTA pD 6.0, 68.75 mM NaCl) and transferred into an NMR tube before being measured on a *Bruker* SUPERSHIELD 600 PLUS with an NMR pulse program developed previously (*Harmel et al., 2019*). $IC_{50}$ curves and values were calculated in *GraphPad* Prism (Version 5.04) using nonlinear regression, where the DMSO control defines 0% inhibition and a no kinase control defines 100% inhibition.

## Kinase-Glo assay optimization

The following solutions were prepared as 4× stock solutions: kinase buffer (HEPES pH 7.4, NaCl, BSA, $MgCl_2$, Tween-20, DTT, and IP6K1$^{L210V}$), DMSO/inhibitor, ADP, and 5PP-InsP$_5$. Final concentrations after optimization were 1 mM ADP, 20 mM HEPES, 50 mM NaCl, 0.2 mg/ml BSA, 2 mM $MgCl_2$, 0.05% Tween-20, 1 mM DTT, 250 nM IP6K1$^{L210V}$, and 50 μM 5PP-InsP$_5$. The reactions were carried out in 384-well plates following this procedure. 5 μl 4× kinase buffer (buffer only for no kinase controls), 5 μl 4× DMSO or inhibitor and 5 μl 4× ADP solution were subsequently added to all wells. After 10 min equilibration, 5 μl 4× 5PP-InsP$_5$ were added to start the reaction (milli-Q water for negative controls). After 3 hr at room temperature, 20 μl of *Promega* Kinase-Glo Plus reagent were added and the luminescence read out with a *Tecan* Infinite M Plex reader using 100-ms exposure time after 10 min of equilibration.

### Z′-plate measurement and high-throughput screen

ADP was prepared in a 2× stock solution in milli-Q water and the pH adjusted to 7.0. The following solutions were prepared as 4× stock solutions: kinase buffer (HEPES pH 7.4, NaCl, BSA, MgCl$_2$, Tween-20, DTT, and IP6K1$^{L210V}$) and 5PP-InsP$_5$. Final concentrations were 1 mM ADP, 20 mM HEPES, 50 mM NaCl, 0.2 mg/ml BSA, 2 mM MgCl$_2$, 0.05% Tween-20, 1 mM DTT, 250 nM IP6K1$^{L210V}$, and 50 µM 5PP-InsP$_5$. The reactions were carried out in 384-well plates following this procedure. 10 µl 2× ADP solution were added to all wells, then 0.2 µl DMSO or compound (1 mM in DMSO) were added followed by the addition of 5 µl 4× kinase buffer. For dose–response curves, compounds were added in serial dilution (twofold serial dilutions in DMSO across multiple plates prior to compound transfer to assay plates, 9 concentrations in total). After 10 min equilibration, 5 µl 4× 5PP-InsP$_5$ were added to start the reaction (milli-Q water for negative controls). After 3 hr at room temperature, 20 µl of *Promega* Kinase-Glo Plus reagent were added and the luminescence read out with a *Tecan* Infinite M Plex or *Perkin Elmer* EnVision reader using 100ms exposure time after 10 min of equilibration. In the counter screen, dose–response curves were measured in the absence of kinase and presence of 50 µM ATP.

### Michaelis–Menten and Lineweaver–Burk kinetics

Kinetic assays were performed in D$_2$O in a total volume of 500 µl containing the same buffer components for IP6K and IP6KA constructs as stated for the NMR activity assays above. However, only 5 mM MgCl$_2$ were added as a constant amount while ATP was added as ATP*Mg solution prepared in twofold dilution starting from 50 mM ATP*Mg was added to final ATP concentrations ranging from 4 mM to 62.6 µM. If applicable, inhibitor was added as a DMSO-$d_6$ stock at a final DMSO concentration of 1%. Otherwise, DMSO-$d_6$ was added to the same percentage. The samples were equilibrated at 37°C for 5 min and the reaction initiated by adding 100 µM [$^{13}$C$_6$]-InsP$_6$. The reaction time was adjusted so that conversion would not exceed 20% to remain in the linear range for initial velocity measurements. The reaction was quenched with 38 µl 700 mM EDTA pH* 8.0 and transferred into an NMR tube before being measured on a *Bruker* SUPERSHIELD 600 PLUS with an NMR pulse program developed previously (*Harmel et al., 2019*). Initial velocities were calculated by dividing the amount of generated product by the product of reaction time in seconds and mass of the kinase in mg. $K_M$ and Lineweaver–Burk plots were generated in GraphPad Prism (Version 5.04.) using nonlinear and linear regression, respectively.

### Protein crystallization and structure determination

The protein construct for crystallization was expressed as follow: An overnight culture of *E. coli Arctic Express* (DE3) harboring pDest-566-MBP-IP6KA$^{M85V}$ (27-270) was diluted to a final OD$_{600}$ of 0.01 in 800 ml TB and grown for 5.5 hr at 37°C. The temperature was switched to 13°C and expression induced with 0.1 mM IPTG after 1 hr. After 20 hr expression, the cells were pelleted by centrifugation (3000 × $g$, 10 min, 4°C) and frozen at −80°C. A week later, the pellet (10 g) was resuspended in 100 ml lysis buffer (50 mM Tris–HCl pH 7.4, 500 mM NaCl, 1 mM MgCl$_2$) supplemented with lysozyme, DNase I, and protease inhibitor. After 30 min of incubation on ice, the cells were homogenized for 30 s and lysed with a microfluidizerTM LM10 at 15,000 psi with three iterations. The cell debris was removed by centrifugation (20,000 × $g$, 20 min, 4°C) and the supernatant lysate filtered (VWR vacuum filter, PES, 0.45 µm). The lysate was adjusted to 50 mM imidazole and loaded onto an equilibrated 5 ml HiTrap IMAC HP column (*GE Healthcare*) at a flowrate of 1.5 ml/min. The column was washed thoroughly with wash buffer (25 mM Tris–HCl pH 7.4, 500 mM NaCl, 50 mM imidazole) and the protein eluted with a 0–100% gradient of elution buffer (25 mM Tris–HCl pH 7.4, 500 mM NaCl, 500 mM imidazole) in wash buffer over 10 CV. Protein-containing fractions were pooled, adjusted to 1 mM β-mercaptoethanol and 0.5 mg TEV protease from (a gift from Martina Leidert) and dialyzed in a 8-kDa cut-off tube overnight against dialysis buffer (50 mM Tris–HCl pH 7.4, 200 mM NaCl, 1 mM DTT). Then, the volume was reduced to 2 ml by spin-filtration through 10 kDa cut-off filters and the TEV-mixture was loaded by injection onto two connected 5 ml MBPTrap HP (GE Healthcare) columns at a flowrate of 2 ml/min. The columns were washed with 3 CV loading buffer (25 mM Tris–HCl pH 7.4, 150 mM NaCl, 1 mM DTT) at a flowrate of 2 ml/min and MBP proteins eluted with 3 CV of elution buffer (25 mM Tris–HCl pH 7.4, 150 mM NaCl, 1 mM DTT, 10 mM D-maltose) at a flowrate of 3 ml/min. Protein-containing fractions from the flow-through were concentrated by spin-filtration through 3 kDa cut-off filters to give 3 ml and again loaded onto two connected 5 ml MBPTrap HP columns at a flowrate of 2 ml/min.

The columns were washed with 3 CV loading buffer at a flowrate of 2 ml/min and MBP eluted with 3 CV of elution buffer at a flowrate of 3 ml/min. Protein-containing fractions from the flow-through were concentrated by spin-filtration through 3 kDa cut-off filters to give 2.5 ml and loaded by injection onto a HiLoad 16/60 Superdex 75 pg column equilibrated with 20 mM Tris–HCl pH 7.4, 150 mM NaCl, and 1 mM DTT. After elution at a flowrate of 0.2 ml/min, protein-containing fractions were concentrated by spin-filtration through 10 kDa cut-off filters to give 300 µl. The concentration of the protein was determined by absorbance spectroscopy at 280 nm using a calculated extinction coefficient of 35,870 $M^{-1}$ $cm^{-1}$. 800 ml culture yielded 11 mg pure protein (17 mg/ml) that was aliquoted to 50 µl aliquots and frozen at −80°C.

The purified *E. histolytica* IP6KA[M85V] (27-270) variant protein (17 mg/ml in 20 mM Tris–HCl pH 7.4, 150 mM NaCl, 1 mM DTT) was complexed with 10 mM $MgCl_2$ and 10 mM ATP, and crystallized using the sitting-drop vapor-diffusion method at 4°C by mixing 300 nl protein and 200 nl reservoir solution (1 mM $MgCl_2$, 460 mM $NaH_2PO_4$). Obtained crystals were soaked for 1 day in 0.1 M sodium acetate pH 5.2, 20 mM $MgCl_2$, 10 mM ATP, and 22% (wt/vol) PEG3350 at 4°C. Before flash-freezing in liquid nitrogen, the crystal was transferred into a cryoprotectant solution consisting of soaking solution supplemented with 33% (vol/vol) ethylene glycol. Diffraction data were collected at 100 K at the beamline BL14.1 operated by the HZB in the BESSY II electron storage ring (Berlin-Adlershof, Germany) (*Mueller et al., 2012*), using a wavelength of 0.9184 Å. Data were processed with the program XDSAPP (*Krug et al., 2012*). The structure was solved by molecular replacement using the program PHASER (*McCoy et al., 2005*) and the known WT *E. histolytica* IP6KA crystal structure with PDB ID code 4o4f as search model (*Wang et al., 2014*). The structure was refined using PHENIX (*Terwilliger et al., 2009*) and the graphics program COOT was used for model building and visualization (*Emsley and Cowtan, 2004*). Figures were created with PYMOL (*Delano, 2002*). Data collection and structure refinement statistics can by found in *Figure 4—figure supplement 1—source data 1*.

## Hydrogen deuterium exchange mass spectrometry

### Deuterium exchange reactions

HDX reactions were performed on the apo enzymes (MBP-*Hs*IP6K1[wt] or MBP-*Hs*IP6K1[L210V]) or enzymes in the presence of inhibitor (FMP-201300). Prior to initiation of deuterium exchange reactions, enzymes were incubated with inhibitor or DMSO blank for 15 min at RT (final concentration: 3 µM enzyme, 50 µM inhibitor, in a buffer consisting of 20 mM HEPES 7.4, 50 mM NaCl, 1% DMSO). From this mixture, 5 µl was taken per sample and 45 µl of deuterated buffer was added to initiate deuterium exchange (final concentrations: 300 nM enzyme, 5 µM inhibitor; final buffer concentration: 20 mM HEPES 7.4, 50 mM NaCl, 85.63% deuterium, 0.1% DMSO). Reactions were carried out in triplicate at 3 (3, 30, and 300 s) or four different time points (0.3, 3, 30, and 300 s) and were quenched by the addition of 20 µl of ice-cold quench buffer (final concentration: 0.57 M GuaHCL, 0.86% formic acid). Samples were snap-frozen in liquid nitrogen and then stored at −80°C until mass analysis.

### Peptide digestion, identification, and measurement of deuterium incorporation

Samples were rapidly thawed and injected onto an ultra-performance liquid chromatography (UPLC) at 2°C. The protein was run over an immobilized pepsin column (Affipro, product number: AP-PC-001), and the peptides were collected onto a pre-column trap (Hypersil GOLD, 2.1 mm × 10 mm, *Thermo Fischer*, product number: 25005-012101). The trap was eluted in line with a an Acquity UPLC BEH C18 column (*Waters*, 130 Å, 1.7 µm, 2.1 mm × 100 mm, product number: 186002352) with a gradient of 10–43% buffer B over 18 min (buffer A 0.1% formic acid, buffer B 99.9% acetonitrile, 0.1% formic acid). Mass spectrometry experiments were performed on an Orbitrap Elite (*Thermo Scientific*) acquiring over a mass range from 308 to 2000 *m/z* using an electrospray ionization source operated at a temperature of 22°C and a spray voltage of 3.8 kV. Peptides were identified using data-dependent acquisition methods following tandem MS/MS experiments. MS/MS datasets were analyzed using MaxQuant against a database of purified proteins and known contaminants with an false discovery rate (FDR) cut-off of 1%.

## Mass analysis of peptide centroids

Deuterium incorporation of peptides was automatically calculated using HDExaminer Software (Sierra Analytics). Peptides were manually inspected for correct parameters (charge state, retention time, isotopic distribution, etc.). Results are presented as relative levels of deuterium incorporation with the only control for back exchange as the deuterium levels in the buffer. Changes in any peptide at any time point greater than specified cut-offs (>6% deuterium incorporation and 0.5 Da between apo and different conditions with an unpaired Student's $t$-test of $p < 0.05$) were considered significant. Source data can be found in *Figure 5—source data 1*.

## Interpretation of results

The results were plotted on a model composed of the AlphaFold structure model of IP6K1 (Q92551), where all affected regions display high (>90%) per-residue confidence scores (pLDDT) (*Jumper et al., 2021*). For orientation, ATP and InsP$_6$ molecules from the crystal structure of *Eh*IP6KA (PDB: 4O4F) were docked in the active site. All conclusions are based on the comprehensively described structural similarities between IP6Ks and protein kinases (*Randall et al., 2020*), the known crystal structure of the ortholog *Eh*IP6KA (*Wang et al., 2014*), and a homology model for mammalian IP6K2 (*Wang et al., 2014*).

### Reverse transcription qPCR

The ribosomal DNA transcription assay was performed like described before (*Morgan et al., 2022*). HCT116 cells (CCL-247 from *ATCC*) were grown to 90–95% confluency before being treated with inhibitor or DMSO for 5 hr (0.1% final DMSO concentration). Total RNA was extracted using the RNeasy kit from *QIAGEN*. cDNA was generated using *Thermo Fisher Scientific* SuperScript III Reverse Transcriptase and 2 μg RNA following the vendor's protocol. Two sets of specific 45S pre-rRNA primers were used as described previously (*Morgan et al., 2022*). The cDNA was diluted 10-fold before conducting qPCR using SYBR Green PCR Master Mix and a StepOnePlus RT PCR System. The difference in transcript levels was calculated using the $\Delta\Delta c_t$ method and technical triplicates.

## Acknowledgements

We thank Steven Moss and Kevan Shokat (UCSF) for providing electrophile-sensitive kinase inhibitors. We further thank Stephen Shears and Huanchen Wang (NIEHS) for the *Eh*IP6KA plasmid and suggestions on *Eh*IP6KA expression and crystallization. We thank Janett Tischer from the Protein Production and Characterization Technology Platform at the Max-Delbrück-Center (MDC) for excellent technical assistance and acknowledge the beamline support by the staff of the Helmholtz-Zentrum Berlin für Materialien und Energie at BESSY. We furthermore thank Han Sun and Haoran Liu (FMP) for discussing docking/MD simulation studies with FMP-201300. We are grateful for helpful discussions and comments from all lab members. Funding: TA was funded by the Deutsche Forschungsgemeinschaft (DFG, German Research Foundation) under Germany´s Excellence Strategy – EXC 2008 – 390540038 – UniSysCat. SH was supported by funding from a DAAD-Leibniz postdoctoral fellowship and from the Swiss National Foundation Sinergia Grant CRSII5_170925. VH and DF acknowledge support by the DFG (TRR186/A08/A24).

## Additional information

### Funding

| Funder | Grant reference number | Author |
|---|---|---|
| Deutsche Forschungsgemeinschaft | EXC 2008 - 390540038 | Tim Aguirre |
| German Academic Exchange Service | DAAD-Leibniz postdoctoral fellowship | Sarah Hostachy |
| Deutsche Forschungsgemeinschaft | TRR186/ A08/ A24 | Volker Haucke Dorothea Fiedler |

| Funder | Grant reference number | Author |
| --- | --- | --- |

The funders had no role in study design, data collection, and interpretation, or the decision to submit the work for publication.

## Author contributions

Tim Aguirre, Conceptualization, Formal analysis, Validation, Investigation, Visualization, Methodology, Writing - original draft, Project administration, Writing – review and editing; Gillian L Dornan, Formal analysis, Investigation, Visualization, Methodology, Writing – review and editing; Sarah Hostachy, Investigation, Methodology, Writing – review and editing; Martin Neuenschwander, Data curation, Formal analysis, Methodology, Writing – review and editing; Carola Seyffarth, Methodology, Writing – review and editing; Volker Haucke, Funding acquisition, Writing – review and editing; Anja Schütz, Formal analysis, Methodology, Writing – review and editing; Jens Peter von Kries, Supervision, Writing – review and editing; Dorothea Fiedler, Conceptualization, Supervision, Funding acquisition, Writing - original draft, Project administration, Writing – review and editing

## Author ORCIDs

Tim Aguirre http://orcid.org/0000-0003-3344-5877
Sarah Hostachy http://orcid.org/0000-0003-1867-9763
Martin Neuenschwander http://orcid.org/0000-0002-3114-7975
Volker Haucke http://orcid.org/0000-0003-3119-6993
Anja Schütz https://orcid.org/0000-0002-0606-2574
Dorothea Fiedler http://orcid.org/0000-0002-0798-946X

Joint Public Review: https://doi.org/10.7554/eLife.88982.3.sa1
Author Response https://doi.org/10.7554/eLife.88982.3.sa2

# Additional files

## Supplementary files

• Supplementary file 1. Hit compounds selective for IP6K1$^{L210V}$. FMP-201300, highlighted in green, is the most promising compound as it is not displaying any PAINS motifs, has no known inhibitory activities, and is neither redox-active nor cytotoxic. All values are for IP6K1$^{L210V}$ if not otherwise indicated.

• Supplementary file 2. Cytotoxicity of selected hits indicated by cell survival after 72 hr incubation at 10 µM concentration. All compounds were measured in independent triplicates and errors represent standard deviation. Redox activity of selected compounds measured via a photometric surrogate assay. Reactions contained 50 mM HEPES pH 7.5, 50 mM NaCl, 200 mM DTT, and 20 µM resazurin, and were incubated for 1 hr at 37°C.

• Supplementary file 3. Primers used for ligation-independent cloning and site-directed mutagenesis.

• MDAR checklist

## Data availability

All data generated or analyzed during this study are included in the manuscript and supporting files; Source Data files have been provided for Figure 4—figure supplement 1 and Figure 5.

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
