## [Editor Report · eLife assessment]

This manuscript describes a **fundamental** strategy for developing isozyme-selective inhibitors of inositol hexakisphosphate kinases. The **compelling** evidence that subtle changes to the gatekeeper position can sensitize the inositol hexakisphosphate kinase mutant to allosteric inhibitors will undoubtedly inspire other analog-sensitive inhibitor studies. This manuscript will be of interest to researchers focusing on kinase regulation and inhibitor design.

---

## [Referee Report · Joint Public Review]

The manuscript by Aguirre et al. describes an elegant approach for developing selective inhibitors of inositol hexakisphosphate kinases (IP6Ks). There are 3 IP6K isozymes (IP6K1-3) in humans, which catalyze the synthesis of inositol pyrophosphates. The lack of isozyme-selective inhibitors has hampered efforts to understand their individual physiological roles. While several inhibitors of IP6Ks have been described, they either lack isozyme selectivity or inhibit other kinases. To address this gap, Aguirre et al. used an analog-sensitive approach, which involves the identification of a mutant that, in an ideal world, doesn't impact the activity of the enzyme but renders it sensitive to an inhibitor that is absolutely selective for the engineered (analog-sensitive) enzyme. Initially, they generated the canonical gatekeeper (Leu210 in IP6K1) mutations (glycine and alanine); unfortunately, these mutations had a deleterious effect on the enzymatic activity of IP6K1. Interestingly, mutation of Leu210 to a valine, a subtly smaller amino acid, didn't affect enzymatic activity. The authors then designed a clever high-throughput assay to identify compounds that show selectivity for L210V IP6K1 versus WT IP6K1. The assay monitors the reverse reaction catalyzed by IP6Ks, monitoring the formation of ATP using a luminescence-based readout. After validating the screen, the authors screened 54,912 compounds. After culling the list of compounds using several criteria, the authors focused on one particular compound, referred to as FMP-201300. FMP-201300 was ~10-fold more potent against L210V IP6K1 compared to WT IP6K1. This selectivity was maintained for IP6K2. Mechanistic studies showed that FMP-201300 is an allosteric inhibitor of IP6K1. The authors also did a small SAR campaign to identify key functional groups required for inhibition.

Overall, this manuscript describes a unique and useful strategy for developing isozyme-selective inhibitors of IP6Ks. The serendipitous finding that subtle changes to the gatekeeper position can sensitize the IP6K1 mutant to allosteric inhibitors will undoubtedly inspire other analog-sensitive inhibitor studies. The manuscript is well-written and the experiments are generally well-controlled.

---

## [Author Response]

The following is the authors’ response to the original reviews.

Please find enclosed our revised manuscript entitled “An unconventional gatekeeper mutation sensitizes inositol hexakisphosphate kinases to an allosteric inhibitor”. We would like to thank the editorial team and the reviewers for carefully reading the manuscript and for raising a number of valuable points. We have included additional data and discussion to address the questions raised.Please find the point-by-point responses below.

**Reviewer #1:**
1. While I understand that FMP-201300 is a tool (proof-of-concept) compound it would be useful to know if it has activity against IP6K1 (or IP6K2) in cells.

We were of course curious about this as well. Unfortunately, our attempts to generate cell lines in which IP6K1 or IP6K2 carry the gatekeeper mutation using CRISPR/Cas editing have not been successful so far. Nevertheless, to obtain information on the permeability and cellular activity of FMP-201300, we decided to treat wt cells, since the compound also inhibited IP6K1-wt and IP6K2wt at higher concentrations.

In a previous study, we could show that reduced intracellular 5PP-InsP5 levels lead to a decrease in rRNA synthesis (https://doi.org/10.1101/2022.11.11.516170). We now repeated this experiment with FMP-201300, along-side the known IP6K inhibitors TNP and SC-919, and could show that FMP-201300 it is able to reproduce this phenotype, strongly suggesting it is capable to diffuse through the cell membrane and act on IP6Ks. We have included this data as a new Figure (Figure S10) and in the discussion part of the manuscript.

1. Did the authors try docking studies to gain insight into the binding site of FMP-201300?

The reviewer raises an important point, and we indeed strongly considered docking studies during the progress of the project. However, given that the HDX-MS data show that the region around the αC-helix becomes much more flexible upon introducing the gatekeeper mutation, we were concerned that docking studies (which would be based on the static wt structure) may not accurately reflect the more dynamic state of the mutated IP6K.

Upon consulting with our colleagues with expertise in docking and molecular dynamics simulations, we believe that MD simulations would need to be performed to obtain a more realistic picture of this protein ligand interaction, which we would like to pursue in the future.

1. Regarding the SAR, it would be useful to know if both carboxylic acids are required for allosteric inhibition.

Given the available data, it appears very likely that both carboxylic acids are required for the inhibitor to unfold its potency. Compound A2, which only contained one carboxylate group, showed drastically reduced potency. We have altered the text in the main manuscript to get this point across more clearly.

1. It would be helpful if the authors presented a model for how they think the Leu210 to Valine mutation sensitizes IP6K1 to FMP-201300.

We agree that it is important to better visualize the structural factors that play a role in the sensitization towards the compound. We have generated a new Figure 5 (and the old Figure 5 is now Supplementary Figure 9), and added a section to demonstrate how we propose the mutation leads to the sensitization of IP6K1 to FMP-201300. For a better understanding, we have also included a depiction how the mutation already affects the apo structures. Furthermore, we have added some text in the HDX section, to better describe the proposed mechanism.

Minor:1. Figure 4: The authors should use the same units in panels a and b.

Thank you for pointing this out, the figure was edited accordingly.

1. In the supplementary Excel file, it would be helpful to include a tab that contains a legend.

A contents page was added to help describe the layout of the supplementary Excel file.

**Reviewer #2:**
Overall, this is an excellent study of high quality. The identified FMP-201300 has the potential for further compound and probe development. My only minor comment is that the authors could spend more time discussing the proposed allosteric binding mode of FMP-201300 and provide more detailed figures to highlight the proposed interactions with the protein and the conformational changes that must ultimately take place to accommodate the allosteric modulator. I appreciate that the co-crystallization experiments did not yield bound inhibitor structures, but perhaps the authors could consider MD simulations to complete their study. However, that could be a story in itself and should not be a must for the publication of this great work.

We agree with the reviewer (and also reviewer 1) that it is important to better visualize the structural factors that play a role in the sensitization towards the compound. We have generated a new Figure 5 (and the old Figure 5 is now Supplementary Figure 9), and added a section to demonstrate how we propose the mutation leads to the sensitization of IP6K1 to FMP-201300. For a better understanding, we have also included a depiction how the mutation already affects the apo structures. Furthermore, we have added some text in the HDX section, to better describe the proposed mechanism. In brief, we propose that the mutation leads to increased flexibility of the region in the mutation, allowing accommodation of FMP-201300 and ATP. These same regions are also the regions that have large decreases in deuterium exchange upon addition of the inhibitor.

We also appreciate the comment about using computational methods, to predict the binding site (also a remark from reviewer 1). We strongly considered docking studies during the progress of the project. However, given that the HDX-MS data show that the region around the αC-helix becomes much more flexible upon introducing the gatekeeper mutation, we were concerned that docking studies (which would be based on the static wt structure) may not accurately reflect the more dynamic state of the mutated IP6K. As the reviewer points out, MD simulations would likely be needed to obtain a more realistic picture of this protein ligand interaction, which we would like to pursue in the future.